# Improved Tolerance of *Lactiplantibacillus plantarum* in the Presence of Acid by the Heterologous Expression of *trxA* from *Oenococcus oeni*

Longxiang Liu [1], Xinyu Yu [1], Ming Wu [2], Keying Zhang [1], Shuai Shang [1], Shuai Peng [3,*] and Weiyu Song [1,*]

[1] Shandong Provincial Engineering and Technology Research Center for Wild Plant Resources Development and Application of Yellow River Delta, College of Biological and Environmental Engineering, Binzhou University, Binzhou 256600, China

[2] Key Laboratory of Quality and Safety of Wolfberry and Wine for State Administration for Market Regulation, Ningxia Food Testing and Research Institute, Yinchuan 750004, China

[3] College of Food Science and Engineering, Gansu Agricultural University, Lanzhou 730070, China

* Correspondence: pengs@gsau.edu.cn (S.P.); songweiyu2007@126.com (W.S.)

**Abstract:** *Oenococcus oeni* is the main microorganism that undergoes malolactic fermentation (MLF) in the winemaking industry due to its excellent adaptability to harsh wine environments. The start of MLF is often delayed or even fails, and low pH appears to be a crucial parameter. To study the function of the *trxA* gene in acid stress, a plasmid containing the *trxA* gene of *O. oeni* SD-2a was heterologously expressed in *Lactiplantibacillus plantarum* WCFS1. The recombinant strain (WCFS1-trxA) grew better than the control strain (WCFS1-Vector) under acid stress. The expression of thioredoxin system genes was much higher in the recombinant strain compared with the control strain under acid stress. In addition, a series of physiological and biochemical assays were conducted. The ATP content was lower in the recombinant strain, while the cell membrane fluidity and integrity improved in the recombinant strain. Moreover, reactive oxygen species (ROS) accumulation, intracellular GSH level, and superoxide dismutase (SOD) activity assays showed that the recombinant strain decreased the intracellular reactive oxygen species (ROS) accumulation by improving the SOD activity. In conclusion, heterologous expression of *trxA* improves the SOD activity of *L. plantarum* WCFS1, reducing bacterial ROS and increasing cell membrane fluidity and integrity, enhancing the tolerance of *Lactiplantibacillus plantarum* WCFS1 under acid stress.

**Keywords:** *Oenococcus oeni*; heterologous expression; thioredoxin system; acid stress; malolactic fermentation

## 1. Introduction

Malolactic fermentation (MLF) is essential in winemaking [1,2]. Successful MLF can convert the malic acid (dicarboxylic acid) into softer-tasting lactic acid (monocarboxylic acid) and carbon dioxide. Thus, the acidity of the wine is reduced, and the microbial stability is improved [3]. In addition, MLF can change the aroma structure of wine, increase the fruity aroma of the wine, and enrich the structure of wine [4]. Therefore, MLF is considered to be a necessary step in the production of high-quality red wine.

The strains that initiate MLF are mainly distributed in *Lactobacillus*, *Lactiplantibacillus*, *Pediococcus*, *Leuconostoc*, and *Oenococcus* [5,6]. Among them, *O. oeni* is the main initiator of MLF [7]. In the process of MLF, the growth and reproduction of these microorganisms will be inhibited by various physical and chemical properties of wine. The four main factors affecting the progress of malolactic fermentation in wine are low pH (3.0–3.5), ethanol (10–16% *v/v*), SO$_2$ (over 10 mg/L), and low temperature (possibly below 12 °C) [8,9].

Currently, there are no efficient gene function research tools for *O. oeni*, such as gene overexpression or gene knockout, which makes the research progress for the *O. oeni* gene function relatively slow [10,11]. More researchers are directing their attention to the use of

*Lactiplantibacillus plantarum* to study the role of exogenous genes, as it is easy to operate genetic transformation technologies in this species. To date, researchers have successfully expressed the *mle*, *hsp*18, *argG*, *puuE*, and *ctsR* genes derived from *O. oeni* in *L. plantarum* and verified the function of these stress-related genes from *O. oeni* using *L. plantarum* [12–16]. Until now, there has been no report on the regulation of the *trxA* gene from *O. oeni* on the stress tolerance to *L. plantarum*.

TrxA is thioredoxin, which is involved in intracellular redox balance, and is induced with mild ethanol stress [17,18]. Margalef-Catala et al. reported that there were three thioredoxin genes (*trxA1*, *trxA2*, and *trxA3*) in the genome of *Oenococcus oeni* PSU-1 [19]. Additionally, the *trxA1* gene *in O. oeni* was horizontally transferred from *Lactobacillus* [19]. Our previous study showed that *O. oeni* SD-2a has only two *trxA* genes [2]. One *trxA* gene was significantly expressed with either acid or ethanol stress pretreatments, and with the increasing intensity of acid or ethanol stresses, the expression level of the *trxA* gene was increasingly overregulated [20]. To investigate the role of this *trxA* under acid stress conditions, the *trxA* gene was amplified from *O. oeni* SD-2a and heterologously expressed in *L. plantarum* WCFS1. The expression of thioredoxin system (Trx system) genes, the growth curve, a series of physiological and biochemical assays including the cell membrane integrity, reactive oxygen species (ROS) accumulation, intracellular ATP and GSH level, and superoxide dismutase (SOD) activity were also determined in the recombinant strain (WCFS1-*trxA*) and the control strain (WCFS1-Vector) under acid stress to investigate the mechanism of *trxA* gene improving acid stress tolerance.

## 2. Materials and Methods

### 2.1. Strains, Growth Conditions, and Plasmids

*O. oeni* SD-2a was screened from the Shandong wine region, preserved at the China General Microbiological Culture Collection Center (CGMCC 0715, Beijing, China), and grown at 28 °C in FMATB medium [19,21]. The cloning host, *Escherichia coli* DH5$\alpha$, was grown at 37 °C in Luria–Bertani medium with 200 µg/mL erythromycin (Solarbio, Beijing, China) when appropriate. *L. plantarum* WCFS1 was grown at 37 °C in de Man–Rogosa–Sharp (MRS) broth [22] and supplemented with 100 µg/mL erythromycin when necessary. Agar plates were also prepared with MRS or LB media containing agar (10 g/L) supplemented with 100 µg/mL erythromycin. All strains and plasmids employed in this study are listed in Table 1.

**Table 1.** Bacterial strains and plasmids used in this study (The superscript lowercase letter 'r' means resistance).

| Strains and Plasmids | Relevant Property | Reference/Source |
|---|---|---|
| *E.coli* DH5$\alpha$ | Cloning host | Takara |
| *O. oeni* SD-2a | Donor bacteria | Our lab |
| *L. plantarum* WCFS1 | Plasmid-free bacteria | Our lab |
| WCFS1(pMG36e) | *L. plantarum* harboring pMG36e, Em [r] | Our lab |
| WCFS1(pMG36e*trxA*) | *L. plantarum* harboring pMG36e*trxA*, Em [r] | This study |
| Plasmids | | |
| pMG36e | *E. coli-L. lactis* shuttle vector (3,6 kb), Em [r] | Our lab |
| pMG36e*trxA* | pMG36e-derivative vector containing the 552 bp region with the *ctsR* gene, Em [r] | This study |

### 2.2. Plasmid Construction and Transformation

The construction of the expression plasmids is illustrated in Figure 1. The *trxA* gene was amplified from genomic DNA using primers 5′-GC<u>GTCGAC</u>AGAGAAGGAG-GAATTATATGGCAAT-3′ and 5′-CC<u>AAGCTT</u>CAGGCTTTTCTTCAATAAAGTATAT-3′; the Hind *III* and Sal *I* (underline) were introduced into the amplified gene. The methods of plasmid construction were adopted from reference [12]. The recombinant plasmid was verified by sequencing and named pMG36e*trxA*; then, it was transformed into *L. plantarum*

WCFS1 by electroporation transformation [12], and these transformants were designated WCFS1-*trxA*.

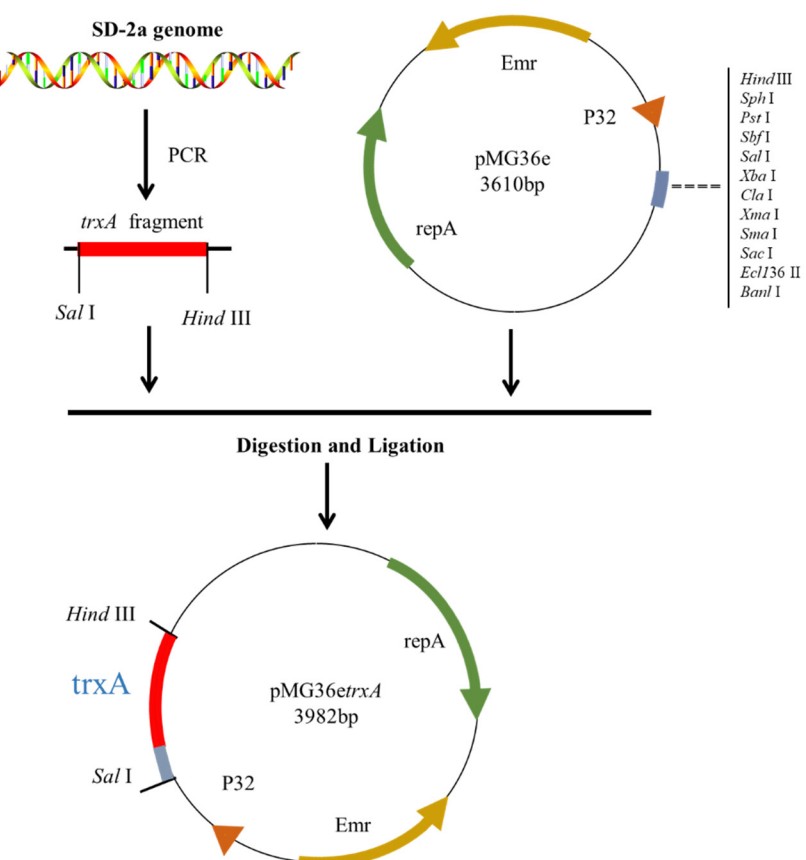

**Figure 1.** Construction of recombinant expression vector pMG36e*trxA*. Emr, erythromycin resistance marker; P32, promoter; repA, replication determinant.

### 2.3. Stress Challenges and the Growth Performance

WCFS1-Vector (*L. plantarum* WCFS1-pMG36e) and WCFS1-*trxA* were cultured overnight in MRS media; then, they were transferred (1%, $v/v$) into fresh MRS media and grown until the $OD_{600nm}$ reached 1.0. To assess the growth performance of both strains, the growth curve was conducted by measuring the absorbance at 600 nm every 4 h under standard MRS (pH 6.3) and acid-stress MRS (from pH 3.2 to pH 4.0, gradient 0.2) medium, with the same inoculum (1%, $v/v$). The pH of the MRS media was adjusted by 1 M HCl using a pH meter (INESA Scientific Instrument Co., Ltd., Shanghai, China), and 100 μg/mL erythromycin was added to all cultures.

### 2.4. ROS Accumulation and Membrane Integrity Evaluation

After analyzing the growth performance of the WCFS1-Vector and WCFS1-trxA under acid stress, pH 3.6 was selected to determine the physiological assays. The WCFS1-Vector and WCFS1-*trxA* cells were collected from an MRS medium (pH 3.6) at the log phase by centrifugation; then, they were washed twice with 10 mM phosphate buffer (PBS, pH 7.0) and resuspended in PBS. The suspension was used to determine the following physiological assays. According to Zhang et al., the oxidant-sensitive probe 2′,7′-dichlorofluorescin diacetate (DCFH-DA) method was adopted to measure the ROS accumulation of strains [23]. The cell suspension was added to 10 μM DCFH-DA and then incubated at 37 °C for 30 min. Fluorescence intensity was measured at $\lambda_{EX}$ 485 nm and $\lambda_{EM}$ 525 nm using a full-wavelength multifunctional microplate reader (Tecan, Männedorf, Switzerland). A positively charged fluorescent nucleic acid dye, propidium iodide (PI), was used to determine the membrane integrity and was used to stain cells with compromised membranes [24].

The samples were collected, washed, and resuspended as previously described; then, they were mixed with 15 μM PI and cultured at 37 °C in the dark for 20 min. Fluorescence intensity was measured at $\lambda_{EX}$ 488 nm and $\lambda_{EM}$ 630 nm using the microplate reader. Values were expressed as the fluorescence intensity per $OD_{600nm}$.

### 2.5. Measurement of Cell Membrane Fluidity

The fluorescence anisotropy was measured with 5 μM DPH (1,6-diphenyl-1,3,5-hexatriene), at $\lambda_{EX}$ 360 nm and $\lambda_{EM}$ 430 nm (5 nm slits). The degree of fluorescence polarization (*p*) and anisotropy (*r*) were calculated according to reference [25].

### 2.6. Intracellular ATP Production, Superoxide Dismutase Activity, and Glutathione Concentration

The ATP Content Assay Kit (Solarbio, Beijing, China) was adopted to measure the intracellular ATP levels according to the manufacturer's instructions. A colorimetric method was employed to measure the total superoxide dismutase (SOD) activity using the Superoxide Dismutase Assay Kit with WST-8 (Beyotime, Shanghai, China). One unit of SOD activity was defined as the amount of enzyme required to decrease 50% of WST-8 formazan formation. The Reduced Glutathione Assay Kit (Nanjing Jiancheng Bioengineering Institute, Nanjing, China) was used to measure the concentration of intracellular glutathione [26]. Additionally, a BCA Protein Assay Kit (Beyotime, Shanghai, China) was used to measure the protein concentration.

### 2.7. Real-Time Quantitative PCR

The Bacteria RNA Extraction Kit (Vazyme Biotech) and HiScript III 1st Strand cDNA Synthesis Kit (Vazyme Biotech) were used to extract RNA and reverse-transcribe it into complementary DNA. Real-time quantitative PCR (RT-qPCR) was performed on a Real-Time PCR Detection System (Thermo Fisher Scientific) using AceQ qPCR SYBR Green Master Mix (Vazyme Biotech). Relative expression levels were calculated by the $2^{-\Delta\Delta Ct}$ method using 16S ribosomal RNA as the reference gene and WCFS1-Vector as the control strain [12]. The sequences of the forward and reverse primers used for analysis are listed in Table S1.

### 2.8. Statistical Analysis

All experiments were performed in triplicate. ROS accumulation, membrane integrity evaluation, intracellular ATP production, SOD activity, GSH concentration, and anisotropy were shown as means with standard deviations (SD). Statistical analysis was performed using the Student's *t*-test with a significance of $p < 0.05$.

## 3. Results and Discussion

### 3.1. Heterologous Expression of pMG36etrxA in L. plantarum WCFS1

The sequences of the trxA fragments were amplified by PCR and then digested with *Sal*I and *Hind*III enzymes. pMG36e plasmid was isolated from *Escherichia coli* DH5α and digested with the same restriction enzymes. Agarose gel electrophoresis was used to test the enzyme digestion effects. As shown in Figure S1a, a linearized DNA band of 3.6 kb indicated that the circular pMG36e plasmids had been digested completely. The band at 0.4 kb represented the digested *trxA* fragments. Agarose gel was cut at the position of the two strands, and the DNA fragments were then purified by a DNA recovery kit. T4 DNA ligase was then used to ligate the two bands. The resulting recombinant plasmid pMG36e*trxA* then formed and was transformed into *E. coli* DH5α. To identify positive clones that contained the resulting recombinant plasmid of pMG36e*trxA*, colony PCR was performed using primers pMG36e-F and pMG36e-R. As shown in Figure S1b, using positive clones as PCR templates can show a band of 1.0 kb in agarose gel (Figure S1b). Meanwhile, the results of the sequencing confirmed that the recombinant plasmid pMG36e*trxA* had successfully been expressed heterologously in DH5α. The pMG36e*trxA* was extracted from DH5α, transformed in WCFS1 by electroporation, and subjected to colony PCR analysis

using the primers pMG36e-F and pMG36e-R to screen recombinant cells. The positive clones showed a band at 1.0 kb (Figure S1c). Meanwhile, the sequencing results confirmed that the recombinant plasmid pMG36e*trxA* had successfully expressed heterologously in WCFS1.

### 3.2. Improved Growth Ability of WCFS1-pMG36etrxA

The effects of heterologously expressed *trxA* on cell growth under acid stress were investigated. As illustrated in Figure 2a, heterologously expressed *trxA* did not influence the growth of *L. plantarum* WCFS1 under normal conditions (MRS media with pH 6.3) compared with the control strain WCFS1-Vector. Contrastingly, the heterologous expression of *trxA* improved cell growth when cells were exposed to acid stress, with a higher maximum $OD_{600\,nm}$ value of WCFS1-*trxA* compared with the WCFS1-Vector (Figure 2b–f). These results demonstrated that the heterologous expression of *trxA* from *O. oeni* SD-2a enhances the tolerance of *L. plantarum* WCFS1 cells under acid stress.

### 3.3. Heterologous Expression of trxA Affects the Transcription Levels of the Inherent Trx System Genes in L. plantarum WCFS1

Since the *trxA* gene often works with other Trx system genes and plays a central role in the Trx system, the expression levels of inherent Trx system genes in WCFS1-*trxA* and WCFS1-Vector were studied under acid stress (pH 3.6). A former study showed that in the *L. plantarum* WCFS1 genome, the Trx system had six ORFs: four thioredoxins genes (*trxA1*, *trxA2*, *trxA3,* and *trxH*), a thioredoxin reductase (*trxB*), and a ferredoxin NAD (P) reductase (*fdr*) [19]. This study calculated the expression levels of all the genes in WCFS1-*trxA* compared with control strain WCFS1-Vector under the same culture conditions. As shown in Figure 3, all Trx system gene expressions were significantly increased. Additionally, the *trxA* gene increased more than 40,000-fold in WCFS1-*trxA* compared with WCFS1-Vector, which indicated that the trxA gene derived from *O. oeni* SD-2a was successfully expressed in WCFS1-*trxA*, while it had almost no expression in the WCFS1-Vector. The expression levels of *fdr*, *trxB*, *trxH*, *trxA3*, *trxA2*, and *trxA1* in WCFS1-trxA increased 12.18-, 15.79-, 40.14-, 64.11-, 75.28-, and 333.27-fold compared with those in the WCFS1-Vector, respectively. These results indicated that the heterologous expression of the *trxA* gene significantly increased the expression level of the inherent thioredoxin system genes in *L. plantarum* WCFS1 cells, conferring these cells with enhanced resistance to damage caused by acid stress. The heterologous expression of the *O. oeni* SD-2a *trxA* gene may affect the activity of the transcription regulator, which activates the transcription of inherent thioredoxin system genes in *L. plantarum* WCFS1. However, the specific mechanism of this phenomenon needs further study.

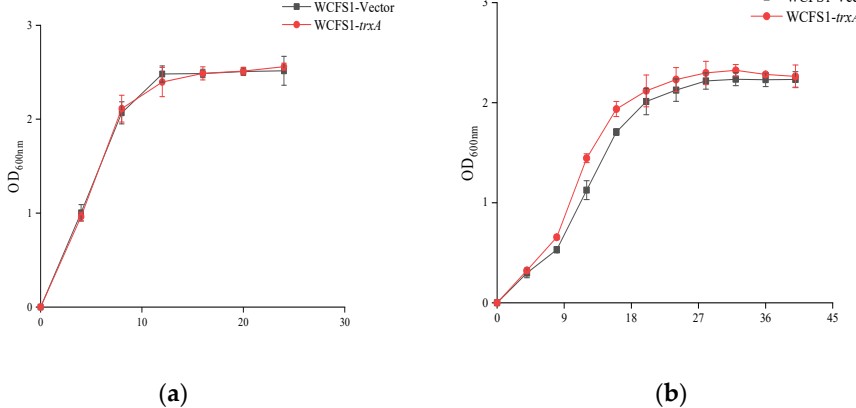

(a)                         (b)

**Figure 2.** *Cont.*

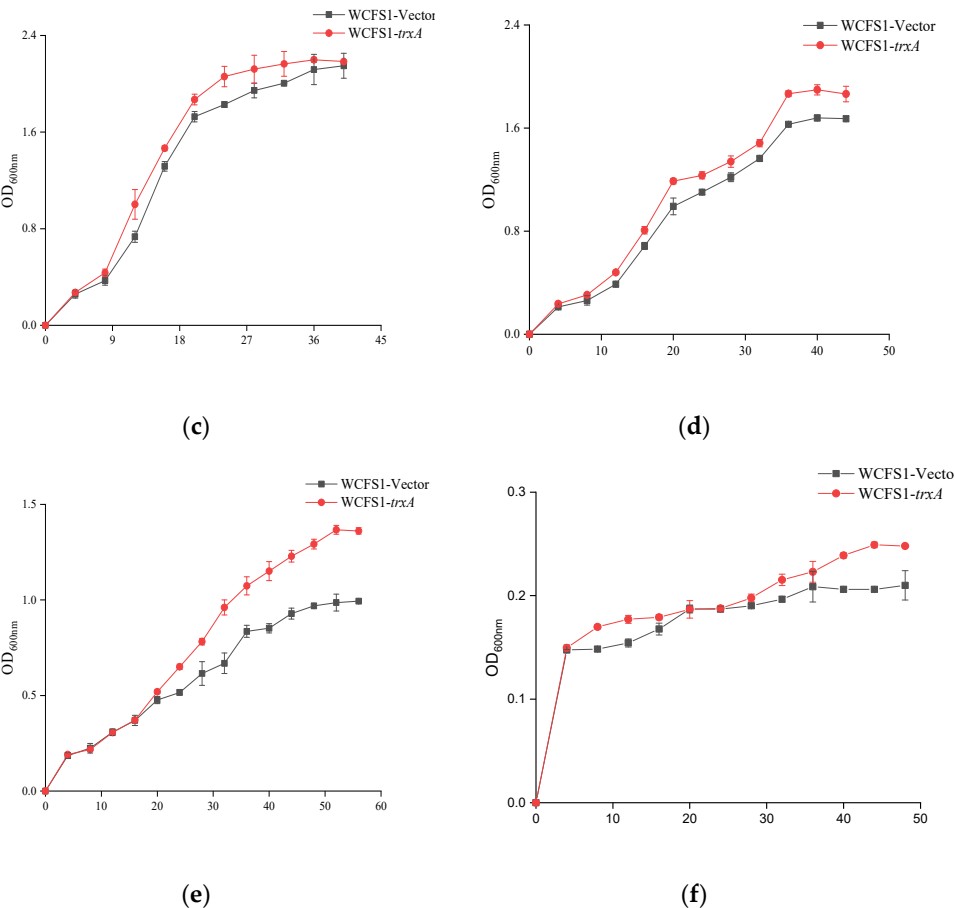

**Figure 2.** Evaluation of the growth performance with strains WCFS1-pMG36e (WCFS1-Vector) and WCFS1-pMG36e*trxA* (WCFS1-*trxA*). The strains were cultured in MRS media with pH 6.3 (**a**), pH 4.0 (**b**), pH 3.8 (**c**), pH 3.6 (**d**), pH 3.4 (**e**), and pH 3.2 (**f**). The number on the *x axis* indicates culture hours. The results are the average of three independent experiments. The T-test of *p* value < 0.05.

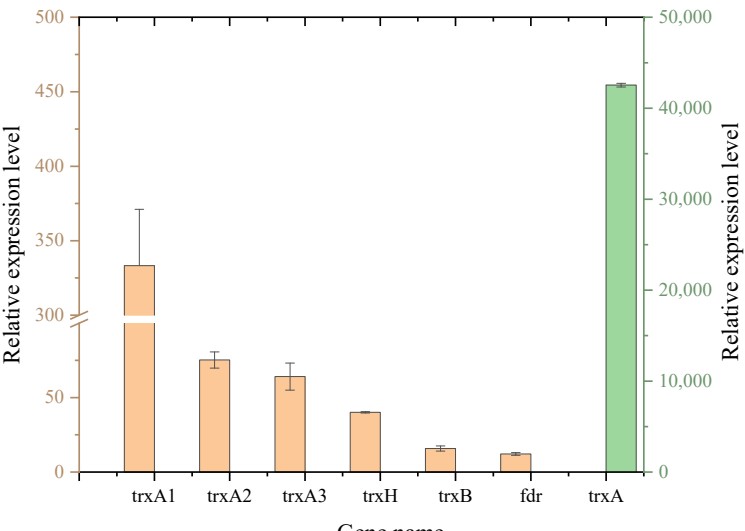

**Figure 3.** The effects of the heterologously expressing *trxA* gene on the transcription of Trx system genes in *L. plantarum* WCFS1. *trxA1, trxA2, trxA3, trxH,* and *trxA* encode thioredoxin A1, thioredoxin A2, thioredoxin A3, thioredoxin H, and thioredoxin A, respectively. *trxB* encodes a thioredoxin reductase. *fdr* is a ferredoxin NAD (P) reductase encoding gene. The difference was significant at a 95% confidence level.

### 3.4. Heterologous Expression of trxA Decreased ATP Content and Increased Cell Membrane Fluidity in L. plantarum WCFS1

As the most important energy source of organisms, ATP plays a vital role in maintaining cell growth, reproduction, and metabolism [27]. It was also reported that cells consume more ATP to maintain growth under stress conditions [28]. As shown in Figure 4, compared with the WCFS1-Vector, the intracellular ATP content of the WCFS1-*trxA* was significantly reduced under the acid stress environment, indicating that *trxA* did not improve the acid stress tolerance of the WCFS1-*trxA* by activating the energy production pathway but accelerated the energy expending response. The extra energy consumed by WCFS1-*trxA* may partially offset the damage caused by acid stress.

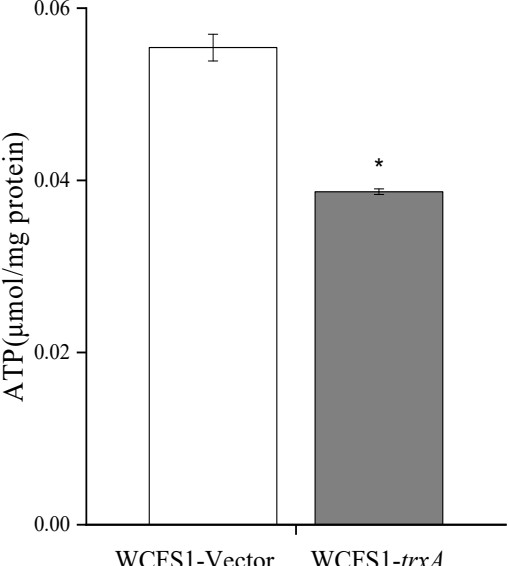

**Figure 4.** The ATP content in the WCFS1-*trxA* and WCFS1-Vector cells cultured in MRS medium (pH 3.6). The ATP content is significantly lower in the WCFS1-*trxA* strain compared with the WCFS1-Vector strain. * means T-test of *p* value < 0.05.

The first barrier of the cell in resisting the external stress environment is the cell membrane [29], and changing the cell membrane fluidity is an essential response of *O. oeni* to external stress [30]. Previous studies have shown that acid stress could reduce the fluidity of cell membranes and cause sclerosis [29,31]. In Figure 5, compared with the control strain, the fluorescence anisotropy of the recombinant strain was significantly lower, and the lower the fluorescence anisotropy, the higher the rotational diffusion rate of the cell membrane. Therefore, the overexpression of the *trxA* gene can increase the rotational diffusion rate of the cell membrane of *L. plantarum* WCFS1 and accelerate the exchange of substances and energy inside and outside the cell.

### 3.5. Heterologous Expression of trxA Results in Decreased ROS Accumulation and Enhanced Cell Membrane Integrity of L. plantarum WCFS1

Acid stress affects the accumulation of intracellular ROS content, causing damage to intracellular components [32] and affecting the integrity of cell membranes [30,31]. In Figure 6, compared with the control strain, the accumulation of ROS in the recombinant strain was significantly lower. The accumulation of ROS would increase the oxidative damage of microorganisms. Therefore, the overexpression of the *trxA* gene can reduce the oxidative damage caused by acid stress to *L. plantarum* WCFS1. The cell membrane integrity assay needs to use a PI probe. The PI probe can stain cells with damaged cell membranes to evaluate the cell membrane integrity of the strain by measuring the fluorescence intensity (FI). The higher the fluorescence value, the worse the cell membrane integrity. Compared with the control strain, the fluorescence value of PI dye in the recombinant strain was

significantly lower, indicating that the cell membrane integrity of the recombinant strain was better. Therefore, overexpression of the *trxA* gene can reduce the damage to the cell membrane of *L. plantarum* WCFS1 caused by acid stress and maintain the integrity of the cell membrane.

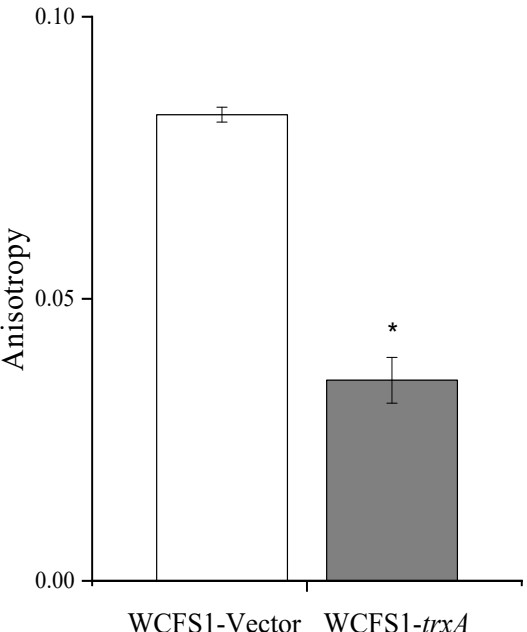

**Figure 5.** The fluorescence anisotropy value of the WCFS1-*trxA* and WCFS1-Vector cells cultured in an MRS medium (pH 3.6). The fluorescence anisotropy value is significantly lower in the WCFS1-*trxA* strain compared with the WCFS1-Vector strain. * means T-test of *p* value < 0.05.

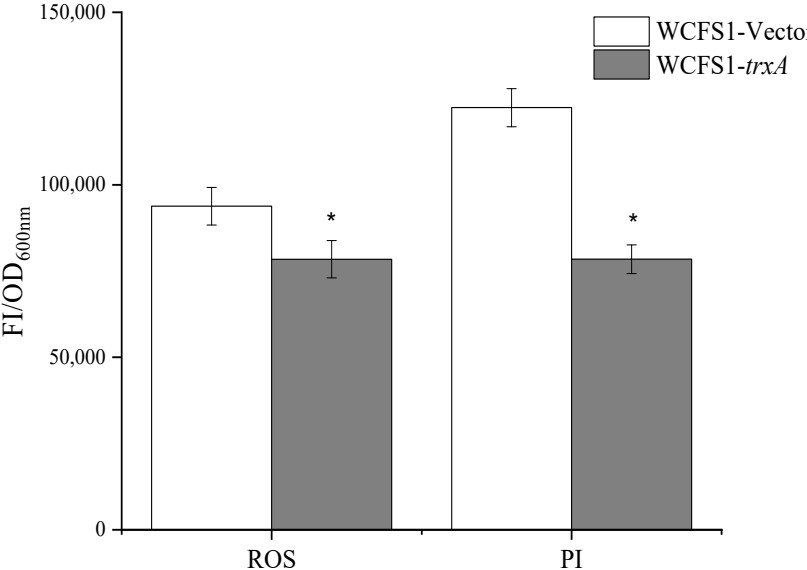

**Figure 6.** The ROS and PI value of the WCFS1-*trxA* and WCFS1-Vector cells cultured in an MRS medium (pH 3.6). The ROS and PI values are significantly lower in the WCFS1-*trxA* strain compared with the WCFS1-Vector strain. * means T-test of *p* value < 0.05.

### 3.6. Heterologous Expression of trxA Decreased ROS Accumulation through Improving SOD Activity

There are two mechanisms (the enzymatic pathway and non-enzymatic pathway) that protect the cells of lactic acid bacteria to reduce the cell damage caused by ROS [19,33]. It can be seen from Figure 6 that the intracellular ROS accumulation of the recombinant

strain was lower compared with the control strain. To explore the reasons for the relatively low ROS content in the recombinant strain, the enzymatic factor SOD activity and the non-enzymatic factor intracellular GSH content were determined. SOD is an important antioxidant that can resist the effect of ROS by eliminating superoxides, while GSH is a non-enzymatic antioxidant with functions such as scavenging free radicals, detoxifying, enhancing immunity, and eliminating the stress produced by ROS. As shown in Figure 7, compared with the control strain, the SOD activity of the recombinant strain was significantly higher. SOD is an essential member of the antioxidant enzyme system in cells, which can help the organism resist oxidative damage. Compared with the control strain, the GSH content of the recombinant strain did not have a significant difference. Therefore, the overexpression of the *trxA* gene mainly removed the accumulated ROS through SOD, thus reducing the oxidative damage caused by acid stress to *L. plantarum* WCFS1.

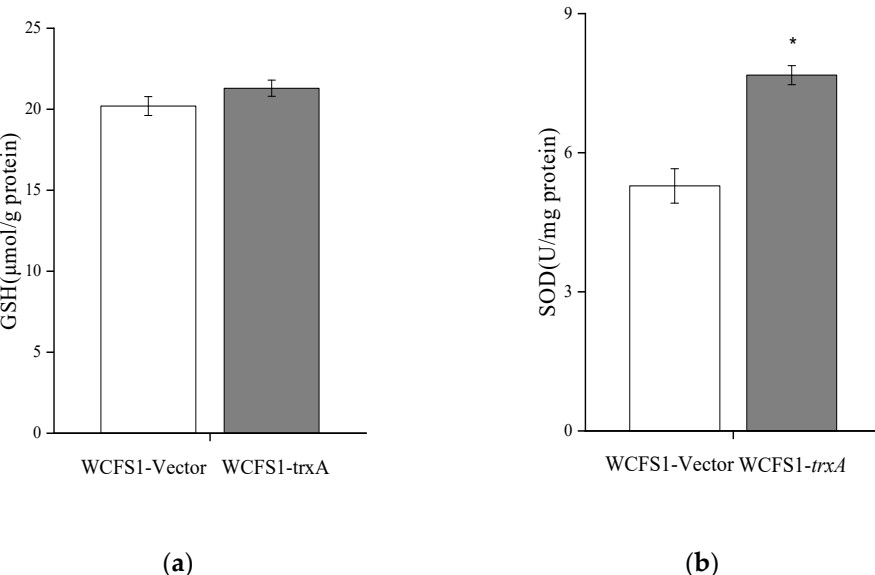

(**a**)　　　　　　　　　　　　　　　　　　　(**b**)

**Figure 7.** The GSH level (**a**) and SOD activity (**b**) of the WCFS1-*trxA* and WCFS1-Vector cells cultured in an MRS medium (pH 3.6). The GSH level is similar in the WCFS1-*trxA* strain compared with the WCFS1-Vector strain, while the SOD activity is significantly higher in the WCFS1-*trxA* strain compared with the WCFS1-Vector strain. * means T-test of *p* value < 0.05.

## 4. Conclusions

The experimental results indicated that the WCFS1-*trxA* showed better growth performance than the WCFS1-Vector under acid stress (pH 3.2, pH 3.4, and pH 3.6). Moreover, the recombinant strain WCFS1-*trxA* showed higher expression of Trx system genes than the control strain under acid stress (pH 3.6), which confirmed that the *trxA* gene indeed regulated the expression of these genes. The *trxA* gene improves acid stress tolerance mainly by improving the SOD activity of *L. plantarum* WCFS1, reducing bacterial ROS, increasing cell membrane fluidity and integrity, and enhancing the acid stress tolerance of *L. plantarum* WCFS1.

**Supplementary Materials:** The following supporting information can be downloaded at: https://www.mdpi.com/article/10.3390/fermentation8090452/s1, Figure S1: The results of agarose gel electrophoresis. Table S1: Primers used for real-Time quantitative PCR.

**Author Contributions:** Conceptualization, L.L., S.P. and W.S.; methodology, L.L., X.Y., M.W. and K.Z.; data curation. S.S.; writing—original draft preparation, L.L. and W.S.; writing—review and editing, L.L., S.S., S.P. and W.S.; funding acquisition, L.L., S.S. and W.S. All authors have read and agreed to the published version of the manuscript.

**Funding:** This research was funded by the National Natural Science Foundation of China (No. 32001659, 32071954), the Shandong Natural Science Foundation of Youth project (ZR2020QC224, ZR2021QD082), the Science and Technology Support Plan for Youth Innovation of Colleges and Universities in Shandong Province (2020KJD005), and the Doctor Start-up Funds of Binzhou University (2020Y05).

**Institutional Review Board Statement:** Not applicable.

**Informed Consent Statement:** Not applicable.

**Data Availability Statement:** Not applicable.

**Conflicts of Interest:** The authors do not report any financial or personal connections with other persons or organizations that might negatively affect the contents of this publication and/or claim authorship rights to this publication.

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
