# Peer review of "Improved Tolerance of Lactiplantibacillus plantarum in the Presence of Acid by the Heterologous Expression of trxA from Oenococcus oeni"

_fermentation, doi:10.3390/fermentation8090452_

Round 1

Reviewer 1 Report

This is an interesting manuscript on the heterologous expression of trxA gene from Oenococcus oeni introduced in Lactiplantibacillus plantarum, which resulted in an improved tolerance of L. plantarum to acid stress. I think there are some important concerning points, as follows.

L14 English usage should be deeply revised along all the manuscript. For instance: L14 ...the main microorganism "undertake" malolactic ....; or L52: ...from O. oeni in L. plantarum, "verification" the function...

L168-172 and Figure 2. Authors do not mention that at pH 3.2 (Figure 2f) the WCFS1-Vector grows better than the -trxA. How can it be explained ?

I think that the improved acid tolerance of this heterologous trxA strain of L. plantarum should be compared experimentally with some O. oeni strains to see if it is better in this aspect when doing the MLF. Have the authors done some experiments in this sense ?

I agree that L. plantarum is an interesting species for the MLF performance in some wines and there are already some commercial starters, but there is a big problem: as you know, there is a strong opposition and legislative blocks to the use of GMO, as this L. plantarum strain, in wine production. Until now, it seems impossible to use GMO in wine industry. So, I would dare to say that he soundness and applied interest of this research is almost null. 

MINOR COMMENTS OR MISTAKES

L50 ".....transformation technologies in this genus". It must be "species" instead of "genus" since it is L. plantarum.

Author Response

Reviewer 1:

This is an interesting manuscript on the heterologous expression of trxA gene from Oenococcus oeni introduced in Lactiplantibacillus plantarum, which resulted in an improved tolerance of L. plantarum to acid stress. I think there are some important concerning points, as follows.

L14 English usage should be deeply revised along all the manuscript. For instance: L14 ...the main microorganism "undertake" malolactic ....; or L52: ...from O. oeni in L. plantarum, "verification" the function...

Response: As required by the reviewer, we have carefully revised the English language along all the manuscript. The modified places are marked in yellow or showing by“Track Changes” function.

L168-172 and Figure 2. Authors do not mention that at pH 3.2 (Figure 2f) the WCFS1-Vector grows better than the -trxA. How can it be explained ?

Response:Thanks very much for the reviewer’s reminder, the mark in Figure 2f is wrong, we have corrected it in the revised manuscript. We are very sorry for the misunderstanding caused by our mistakes.

I think that the improved acid tolerance of this heterologous trxA strain of L. plantarum should be compared experimentally with some O. oeni strains to see if it is better in this aspect when doing the MLF. Have the authors done some experiments in this sense ?

Response:This is a very interesting suggestion. We did not make such a comparison. In our further study we can do some experiments in this sense and this maybe another interesting story.

I agree that L. plantarum is an interesting species for the MLF performance in some wines and there are already some commercial starters, but there is a big problem: as you know, there is a strong opposition and legislative blocks to the use of GMO, as this L. plantarum strain, in wine production. Until now, it seems impossible to use GMO in wine industry. So, I would dare to say that he soundness and applied interest of this research is almost null. 

Response: It is very common to use transgenic technology to improve the stress resistance of species. It is also significant to understand the mechanism of stress resistance enhancement of transgenic organisms. In addition to our research, Schumann et al. (Heterologous expression of Oenococcus oeni malolactic enzyme in Lactobacillus plantarum for improved malolactic fermentation. AMB Express, 2012, 2, (1), 19), Weidmann et al. (Production of the small heat shock protein Lo18 from Oenococcus oeni in Lactococcus lactis improves its stress tolerance. Int. J Food Microbiol.2017, 247, 18-23), Yuan et al. (Heterologous expression of the puuE from Oenococcus oeni SD-2a in Lactobacillus plantarum WCFS1 improves ethanol tolerance. J Basic Microbiol. 2019, 59, (11), 1134-1142), Zhao et al. (Heterologous expression of argininosuccinate synthase from Oenococcus oeni enhances the acid resistance of Lactobacillus plantarum. Front. Microbiol. 2019, 10, 1393; Heterologous expression of ctsR from Oenococcus oeni enhances the acid-ethanol resistance of Lactobacillus plantarum. FEMS Microbiol. Lett. 2019, 366) used similar methods to study the performance improvement mechanism of transgenic organisms in certain aspects. The legislative blocks to the use of GMO may be not immutable and these studies lay the foundation for future applications.

MINOR COMMENTS OR MISTAKES

L50 ".....transformation technologies in this genus". It must be "species" instead of "genus" since it is L. plantarum.

Response: Thanks very much for the reviewer’s reminder, we have corrected this in the revised manuscript in line 52.

Reviewer 2 Report

Please supplement trx A protein expression test results, such as SDS-PAGE or WB.

Author Response

Reviewer 2:

Please supplement trxA protein expression test results, such as SDS-PAGE or WB.

Response: We didn’t test Trx A protein expression partially for that we do not have antibodies to TrxA. Besides, the expression vector pMG36e we used in this study had successfully expressed some other genes (like argG, ctsR and puuE) in Lactiplantibacillus plantarum (Heterologous expression of argininosuccinate synthase from Oenococcus oeni enhances the acid resistance of Lactobacillus plantarum. Front. Microbiol. 2019, 10, 1393; Heterologous expression of ctsR from Oenococcus oeni enhances the acid-ethanol resistance of Lactobacillus plantarum. FEMS Microbiol. Lett. 2019, 366; Heterologous expression of the puuE from Oenococcus oeni SD-2a in Lactobacillus plantarum WCFS1 improves ethanol tolerance. J Basic Microbiol. 2019, 59, (11), 1134-1142). And in these previous studies, RT-qPCR was also used to detect gene expression. In our research, we test the expression level of trxA and found it successfully expressed in the recombinant strain. Combined with the phenotypic data of the the recombinant strain, we have reason to believe that the Trx A protein had successfully expressed in the recombinant strain.

Reviewer 3 Report

The study “Improved tolerance of Lactiplantibacillus plantarum in the 2 presence of acid by heterologous expression of trxA from Oeno- 3 coccus oeni” is well designed, well written and the results are vital to be published. The authors  showed that the recombinant strain WCFS1-trxA showed better growth performance under acid stress and showed higher expression of Trx system genes compared with the control strain indicating trxA gene  regulated the expression of these genes.  However, I have some minor corrections,

L105  show the author names after (according to…)

Fig 3 add title of x axis and clarify all abbreviations in the legend

Fig 4,5,6,7 clarify x-axis title

L283-284, check the format Pichia fermentans

L328,329, 348 check the format of lactobacilli,  cervisiae, and Enteritidis

Author Response

Reviewer 3:

The study “Improved tolerance of Lactiplantibacillus plantarum in the 2 presence of acid by heterologous expression of trxA from Oeno- 3 coccus oeni” is well designed, well written and the results are vital to be published. The authors  showed that the recombinant strain WCFS1-trxAshowed better growth performance under acid stress and showed higher expression of Trx system genes compared with the control strain indicating trxA gene  regulated the expression of these genes.  However, I have some minor corrections,

L105  show the author names after (according to…)

Response: We have showed the author names in the revised manuscript in line 113.

Fig 3 add title of x axis and clarify all abbreviations in the legend

Response: We have done this in the the revised manuscript in line 242-246.

Fig 4,5,6,7 clarify x-axis title

Response:We have done this in the the revised manuscript in line 260-261, 273-274, 293-294, 314-316.

L283-284, check the format Pichia fermentans

Response: We have corrected it in the the revised manuscript in line 335-336.

L328,329, 348 check the format of lactobacilli,  cervisiae, and Enteritidis

Response: We have corrected it in the the revised manuscript in line 382, 383, 402.

Round 2

Reviewer 1 Report

Authors have answered all my concerns and have improved substantially the manuscript.

Nevertheless, English usage is still not good. For instance, just beginning the Abstract, line 14: "Oenococcus oeni is the main microorganism undertakes malolactic fermentation..." You are using two present tense verbs in the same sentence: "is" and "undertakes"

As said, I think these results should be compared with O. oeni strains in order to see if this heterologous expression of L. plantarum is better or not.

As you mention, it is certain that there are some other scientific works of transgenic technology of O. oeni genes in other LAB, but with the current blocking legislation on wine GMO, the interest and possible apllication of this subject is very low.

Author Response

Authors have answered all my concerns and have improved substantially the manuscript.

Nevertheless, English usage is still not good. For instance, just beginning the Abstract, line 14: "Oenococcus oeni is the main microorganism undertakes malolactic fermentation..." You are using two present tense verbs in the same sentence: "is" and "undertakes"

As said, I think these results should be compared with O. oeni strains in order to see if this heterologous expression of L. plantarum is better or not.

As you mention, it is certain that there are some other scientific works of transgenic technology of O. oeni genes in other LAB, but with the current blocking legislation on wine GMO, the interest and possible apllication of this subject is very low.

Response: We have submitted our manuscript to MDPI for English editing and the revisions made to the manuscript had been marked up using the “Track Changes” function. In our further study, we will do some experiments to compare the MLF performance of O. oeni strains with heterologous expression of L. plantarum. Our group also try to use directed evolution method to generate superior O. oeni strains that can conduct more efficient MLF, and this method maybe a better way to enhance the commercial application value of LAB strains.

Reviewer 2 Report

I agreed to accept

Author Response

Thanks the reviewer.